# Peripheral Nerve Regeneration at 1 Year: Biodegradable Polybutylene Succinate Artificial Scaffold vs. Conventional Epineurial Sutures

**DOI:** 10.3390/polym15163398

**Published:** 2023-08-14

**Authors:** Luca Cicero, Roberto Puleio, Giovanni Cassata, Roberta Cirincione, Lawrence Camarda, Dario Caracappa, Lorenzo D’Itri, Mariano Licciardi, Giulio Edoardo Vigni

**Affiliations:** 1Centro Mediterraneo Ricerca e Training (Ce.Me.Ri.T), Istituto Zooprofilattico Sperimentale della Sicilia “A. Mirri”, 90129 Palermo, Italy; lucacicero031182@gmail.com (L.C.); giovanni.cassata@izssicilia.it (G.C.); robertacirincione@gmail.com (R.C.); 2Laboratorio Istopatologia e Immunoistochimica, Dipartimento Ricerca Biotecnologica e Diagnostica Specialistica, Istituto Zooprofilattico Sperimentale della Sicilia “A. Mirri”, 90129 Palermo, Italy; roberto.puleio@izssicilia.it; 3Department of Orthopaedics and Traumatology, University of Palermo, 90133 Palermo, Italy; lawrence.camarda@unipa.it (L.C.); dr.ditri@gmail.com (L.D.); giulio.vigni@gmail.com (G.E.V.); 4Dipartimento di Discipline Chirurgiche, Oncologiche e Stomatologiche (DICHIRONS), Università degli Studi di Palermo, 90127 Palermo, Italy; dario.caracappa@gmail.com; 5Dipartimento di Scienze e Tecnologie Biologiche Chimiche e Farmaceutiche (STEBICEF), Università degli Studi di Palermo, 90132 Palermo, Italy

**Keywords:** poly(1,4-butylene succinate), nerve regeneration, electrospinning

## Abstract

The utilization of a planar poly(1,4-butylene succinate) (PBS) scaffold has been demonstrated as an effective approach for preserving nerve continuity and facilitating nerve regeneration. In this study, we assessed the characteristics of a microfibrous tubular scaffold specifically designed and fabricated through electrospinning, utilizing PBS as a biocompatible and biodegradable material. These scaffolds were evaluated as nerve guide conduits in a rat model of sciatic nerve neurotmesis, demonstrating both their biodegradability and efficacy in enhancing the reconstruction process over a long-term period (1-year follow-up). Histological assay and electrophysiological evaluation were performed to compare the long-term outcomes following sutureless repair with the microfibrillar wrap to outcomes obtained using traditional suture repair.

## 1. Introduction

Peripheral nerve injuries (PNI) can significantly impact quality of life, regardless of their etiology. Despite the availability of multiple treatments, achieving complete functional recovery remains challenging [1]. Insufficient nerve recovery leads to muscle atrophy, chronic pain, and profound weakness [2]. Moreover, significant loss of nerve tissue and prolonged denervation of proximal nerves increase the likelihood of irreversible atrophy in innervated organs [3]. Consequently, time becomes a critical factor as the regenerative process is inherently time-consuming, particularly in the absence of external intervention [1]. In fact, long-distance axonal regeneration occurs at a slow rate of 1–3 mm per day to reach and reinnervate distal motor endplates [4]. Relying solely on spontaneous recovery poses risks, as there is a window of opportunity during which surgery can substantially enhance the potential for recovery [5].

Treatment options generally involve microsurgical techniques such as direct repair, tension-free end-to-end sutures, and the use of autologous nerve grafts for repairing larger gaps [6,7,8]. While autografts remain the gold standard, they have inherent limitations including donor site morbidity and limited availability. Consequently, recent studies have focused on developing new methods to promote axonal regeneration, including pharmacological treatments, cell-based therapies, growth factors, gene therapies, and surgeries combined with the implantation of various biomaterials [1,5,9,10,11,12,13,14].

Among these approaches, biomimetic electrospun nanofiber scaffolds have gained considerable attention in the field of tissue engineering and are commonly employed for nerve tissue engineering [11,12]. These structures not only guide nerve regeneration but can also incorporate support cells, genetic manipulators, and growth factors, and enable electroconduction [1,15].

The use of planar poly(1,4-butylene succinate) (PBS) scaffolds has been demonstrated as an effective method for preserving nerve continuity and promoting nerve regeneration [16]. In a recent study, Miceli et al. evaluated the properties of microfibrous tubular scaffolds designed and fabricated through electrospinning, using PBS as a biocompatible and biodegradable material. The optimized scaffold morphology featured a small diameter and micro-porous conduit, facilitating cell integration, adhesion, and growth while preventing cell infiltration through the graft’s wall. Mechanical properties of the tubular conduits fell within the physiological range as well [17]. Additionally, Vigni et al. demonstrated in another study that a PBS microfibrillar scaffold used in critical bone defects on a rabbit model could potentially enhance bone regeneration [18].

This combination of characteristics highlights the versatility of PBS as a biomaterial for producing scaffolds suitable for various biomedical applications. Building upon the promising results observed within a few months of implantation, the objective of this study is to confirm its long-term effectiveness.

## 2. Materials and Methods

### 2.1. Poly-Butylene Succinate (PBS) Scaffolds Fabrication by Electrospinning Technique

PBS scaffold was produced following the procedure previously published [16]. Briefly, a PBS (Poly(1,4-butylene succinate) extended with 1,6-diisocyanatohexane, Tm 120 °C, Aldrich, UK) solution (30 mL) in dichloromethane (15% *w*/*v*) was used to prepare each batch of the scaffold. The electrospinning process was carried out horizontally with 15 kV voltage (Spellman CZE 1000 R, Hauppauge, NY, USA) and a constant polymeric solution rate (0.8 mL/min) obtained through a programmable syringe pump (Aitecs PLUS SEP-21, Vilnius, Lithuania). The electrospun scaffold was collected on a stainless steel earthed rotating collector, positioned 15–20 cm away from the tip of the needle.

Molecular weight evaluation of PBS was carried out via size exclusion chromatography (SEC) by using an Agilent 1260 Infinity multi-detector GPC/SEC system. The elution was performed on a Phenogel 10^4^ column (Phenomenex, Torrance, CA, USA) using Hexafluoroisopropanol (Sigma-Aldrich, Gillingham, UK) as a mobile phase at 30 °C, with a flow rate of 0.6 mL/min. Standards of PEG were used for calibration. The average molecular weight (Mw) and polydispersity (Mw/Mn) resulted 294,500 and 1.4, respectively.

### 2.2. PBS Scaffolds Characterization by Scanning Electron Microscopy (SEM) and Microcomputed Tomography (μCT)

Scanning electron microscopy was carried out using an ESEM Philips XL30 microscope, operating at 5 kV. Each sample was deposited onto a carbon-coated steel stub, dried under vacuum (0.1 Torr), and sputter-coated with gold (15 nm thickness) prior to microscopy examination.

MicroCT scanner Skyscan 1272, Bruker, Kontich, Belgium was used to analyze the 3D structure of the scaffold at a source voltage of 40 kV, a current of 250 mA, a total rotation of 180°, and a rotation step of 0.3°. No filter mode was chosen for the acquisitions. The image pixel size was 2.6 μm and the scan duration was about 3 h for every sample. The scanning dataset obtained after the acquisition step consisted of images in 16-bit tiff format (3238 × 4904 pixels). The 3D reconstructions were carried out using the software NRecon (version 1.6.10.2), starting from the acquired projection images. The obtained 2D-images had a color depth of 8 bit with 265 grey levels. The whole set of raw images were displayed in a 3D space with the software CTVox.

### 2.3. Animals

All experiments were performed in the Istituto Zooprofilattico Sperimentale della Sicilia “A. Mirri” (Palermo, Italy) and authorized by the Ministry of Health (Rome, Italy; Authorization Number 456/2018-PR). Procedures involving animals were carried out in accordance with the Italian Legislative Decree N° 26/2014 and the European Directive 2010/63/UE. Ten adults male Wistar rats, weighting between 150 and 200 g (Charles River Laboratories, Calco, Italy), were used for this study. Animals were housed in polypropylene cages and kept in controlled temperature (22 ± 2 °C), humidity (50–55%), and light (12 h light/dark cycle). Animals had access to food and water ad libitum. Rats were allowed to acclimate for at least 2 days prior to experiments.

Ten adults male Wistar rats were subjected to surgical neurotmesis of the right sciatic nerve and randomly divided into two experimental groups. In Group 1 (G1; Control; n = 5), conventional epineural microsurgical sutures were used to repair the PNI. In Group 2 (G2; Microfiber wrap; n = 5), a planar microfibrillar scaffold (composed of Poly(1,4-butylene succinate) extended with 1,6-diisocyanatohexane (PBS) was implanted and wrapped around the nerve ends separated by a 7 mm gap, without any epineural repair.

### 2.4. Surgical Procedure and Scaffold Implantation

Surgical procedures were performed under aseptic conditions using a power focus surgical microscope (Carl Zeiss, Oberkochen, Germany). Animals were induced to anesthetic depth with inhaled isoflurane at 2% and then anesthetized with intramuscular (i.m.) injection of Zoletil(r) (tiletamine/zolazepam; 10 mg/kg) and Domitor(r) (medetomidine hydrochloride; 0.5 mg/kg) [18]. All rats were operated on by the same surgeon. Each animal underwent surgery only on the right limb, so that mobility and self-sufficiency in eating and drinking were preserved. Before surgery, hair was clipped over the thigh and the surgical field was scrubbed with a 70% alcohol solution. A skin incision of 40 mm was performed over the right gluteal muscle along the femoral axis in each rat. Proceeding with blunt dissection, the biceps femoris and the superficial gluteal muscles were retracted. The sciatic nerve was exposed and then sharply interrupted using micro-scissors at the mid-thigh level, proximal to the tibial and peroneal bifurcation. In the control group (G1), the proximal and distal nerve ends of the injured nerve were sutured using three 6/0 monofilament nylon epineural sutures (Ethicon). In the nanofiber wrap group (G2) no primary repair was performed. After the nerve section, a 7 mm long gap was created. This simulated a “facilitated” nerve retraction. The proximal and distal nerve ends were wrapped in the PBS nanofiber scaffold (12 × 12 mm) for 2.5 mm along with the 7 mm gap area. The resultant wrapped scaffold was 1–1.5 mm larger than the nerve diameter. The sciatic nerve was kept moist with sterile saline solution throughout the surgical procedure. The skin incision was closed with reabsorbable sutures and disinfected with povidone–iodine (Betadine) solution. I.m. atipamezole (Antisedan) (300 μg/kg) was used in order to awaken all rats. Carprofen (5 mg/kg) and Enrofloxacin (5 mg/kg) were administered daily for 1 week to each rat. After the procedure, each animal was assigned with an identification number and housed one per cage. They were monitored on a daily basis for infection, self-mutilation, and signs of distress.

### 2.5. Histological Analysis

The histological analysis was performed at 1 year. Using a surgical microscope, 10 mm of sciatic nerve was removed from each animal at the same anatomic location (distal to the PNI and 5 mm distal to where the sciatic nerve crosses the tendon of the internal obturator muscle). Both sciatic nerves were harvested from each animal: the native left side and the treated right side. Nerve samples were immediately fixed in 4% paraformaldehyde in phosphate-buffered (PB) saline for 2–4 h, then washed and stored in 0.2% glycine in PB saline. Afterwards, the specimens were washed with PB saline, post-fixed with 2% osmium tetroxide for 2 h, washed with 3–5 passages in distilled water, dehydrated with an increasing alcohol series, and embedded in paraffin. Finally, the specimens were cut into transverse thin sections (3–5 μm thick) and stained with hematoxylin and eosin for morphometric analyses [19]. One more sciatic nerve aliquot (5 mm proximal to the PNI) was directly stained with hematoxylin and eosin for histological evaluation. Each specimen was assigned with an identification number without any reference to the groups by LC. Then, the specimens were evaluated by a single researcher (RP) for overall nerve architecture, quantity of regenerated nerve fibers, and Wallerian degeneration. All nerve sections were evaluated under an optical microscope (Leica DMR, Wetzlar, Germany) and photographed with a high-resolution camera (Nikon DS-Fi1, Tokyo, Japan). The sciatic nerve area was calculated for each experimental group at follow-up. Six random microscopic fields per nerve were captured at 1000× magnification and evaluated with image analysis software (Image J, Version 1.53t), based on gray and white scales. Myelinated fibers were semi-automatically recognized by the software and the remaining fibers were manually redrawn. Total fiber number (N) was estimated by measuring sciatic nerve area and the area of sample at 1000× magnification, multiplying by the number of fibers in sample. The fiber density (FD = N/mm^2^) was calculated by dividing the number of fibers within the sampling field by its area [20].

### 2.6. Compound Muscle Action Potential (CMAP) Response Evaluation

The registration of the compound muscle action potential (CMAP) was performed using a pair of monopolar needle electrodes applied in a belly-tendon scheme. The reference electrode was inserted into the subcutaneous tissue of one of the front limbs. Stimulation was achieved through a single pulse of 0.1 ms duration using a biphasic square wave; the amplitude was measured from peak to peak.

Subsequently, the motor unit number estimation (MUNE) technique was used to estimate the number of motor units comprising a muscle. This was obtained by dividing the CMAP amplitude by the average amplitude of the individual single motor unit action potentials (SMUAP).

### 2.7. Statistical Analysis

All results are reported as mean ± standard deviation, and statistical analysis for significance was performed by means of Student’s *t*-test, using GraphPad PrismTM 4.0 software (GraphPad Software Inc., San Diego, CA, USA), assuming unequal variance and two-tailed distribution; values of *p* < 0.05 were considered statistically significant.

## 3. Results and Discussion

### 3.1. Fabrication and Characterization of PBS Scaffolds

The PBS scaffold tested in this study was produced via electrospinning using the procedure already published [16]. In order to identify the characteristics of the implanted material, SEM and microCT images are herein reported (Figure 1).

The above analysis demonstrated the particular micro-porosity on the fibers surface (yellow arrows in Figure 1) and the micro-fibrillar structure of the scaffold, showing microfibers with a diameter between 1–5 microns, alternating with the presence of collapsed-balloons like structures along the microfibers (red arrows in Figure 1). Micro-CT 3D reconstruction show the typical planar shaper of the scaffold, its thickness (300 microns), and low density of the material.

### 3.2. Evaluation of Nerve Regeneration by Histological Analysis: Counting of Regenerated Fibers

Peripheral nerve injuries (PNI) are neurological disorders that significantly impact patients’ daily functions and routines, leading to severe and lifelong functional and physiological disabilities. Approximately 33% of all PNI cases result in poor functional outcomes, incomplete nerve recovery, and the loss of sensory and motor function. The extensive loss of nerve tissue and prolonged denervation of proximal nerves increase the risk of irreversible atrophy in muscles innervated by the affected nerves. Current management of PNI primarily involves microsurgical approaches such as direct repair, tension-free end-to-end suturing, and the gold standard method of using autologous nerve grafts for bridging large nerve gap (>3 cm), critical nerve injuries, and more proximal damages [1]. Nerve guide conduits serve as a viable surgical alternative to autografts, addressing many of their limitations. Through the use of natural or synthetic biopolymers, tissue engineering has developed tubular structures that function as a bridge between the proximal and distal ends of the injured nerve, supporting and facilitating axonal regrowth within the conduit and promoting infiltration of surrounding tissues. A significant advantage of conduit implantation is its ability to create an optimal microenvironment for neuronal recovery while reducing the risk of perineural fibrosis, neuroma formation, and inflammation. To achieve successful peripheral nerve repair, an ideal conduit should possess various properties, including biomimetic architecture, appropriate wall thickness, low toxicity, trophic support permeability, neuroinductivity, neuroconductivity, biodegradability, biocompatibility, and flexibility.

Planar poly(1,4-butylene succinate) (PBS)-based scaffolds have already demonstrated excellent efficacy as implantable three-dimensional biomaterials in various applications of regenerative medicine, including peripheral nerve [16] and bone [18] regeneration. The effectiveness of the discussed scaffold in guiding and enhancing nerve functional recovery was demonstrated in a rat model of sciatic nerve transection (Figure 2). This study confirmed its complete biodegradability and reabsorption without causing inflammation, physiological complications, or rejection, even after 120 days post-implantation [16].

Furthermore, recent study results have demonstrated the osteointegration capability of the biodegradable PBS scaffold and its ability to support and enhance bone tissue formation when used for repairing bone defects in an in vivo rabbit model [18]. Building upon these promising outcomes observed at 4 months post-surgery, this study presents the successful utilization of the planar PBS scaffold in an in vivo rat model of sciatic nerve neurotmesis, showcasing its biodegradability and effectiveness in improving the reconstruction process over a long-term period (1-year follow-up). To evaluate the in vivo biocompatibility and effectiveness of the PBS scaffold as a nerve guide conduit, histological evaluations were conducted 1 year after implantation. The results of the suture-less repair with the microfibrillar scaffold (Group 2) were compared to those obtained with the traditional suture repair (Group 1).

Examination of the normal sciatic nerve using hematoxylin-eosin stain revealed the typical undulated and parallel organization of the nerve fibers (red arrows in Figure 3A) with few but large fibers, compared to Group 2 (Figure 3B) with many and small fibers. In particular, nerve fibers of Group 2 (black arrows in Figure 3B) show a great number of axonal gems that, organized in growth cones, generated new reinnervation (higher fiber density).

At 1 year after surgery, the total fiber number (Figure 4A) and fiber density (Figure 4B) at the injury site were evaluated for animals in Groups 1 and 2. Histological analysis demonstrated that the nanofiber wrap group exhibited a significantly higher total number of fibers compared to the conventional suture repair group (9867 ± 674.326 vs. 7890 ± 535.52) (Figure 4A). The fiber density (number of fibers/mm^2^) showed a statistically significant difference between Groups 1 and 2 at the 12-month post-surgery mark (Figure 4B). In Group 1, the fiber density was 12361 ± 651.83 fibers/mm^2^, while an increase was observed in Group 2, the nanofiber wrap group (14683 ± 602.96 fibers/mm^2^). Histological findings confirmed the absence of inflammatory cell infiltration, Wallerian degeneration, and perineural fibrosis at the repair site in animals treated with the PBS scaffold.

### 3.3. Electrophysiological Evaluation

The analysis of the *t*-test for independent variables was performed on the motor unit number estimation (MUNE) of each gastrocnemius and tibialis anterior muscle, with the variable “side” (right vs. left) as the within-subject factor. At the 12-month follow-up, statistical significance (*p* > 0.05) in the mean MUNE (estimated number of motor units) was investigated, both in the operated limbs (right, R) and the healthy ones (left, L), as well as for the gastrocnemius (GM) and the tibialis anterior (TA), considering the variable “side” (right vs. left) as the within-subject factor.

The experimental setting is shown if Figure 5.

The analysis of the gastrocnemius muscle (GM) showed no significant difference (*p* > 0.05) only in the scaffold group (GM-R vs. GM-L, 23.13 vs. 42.89). However, statistical significance was observed in the control group (GM-R vs. GM-L, 19.85 vs. 56.85) (Figure 6).

Regarding the tibialis anterior muscle (TA) analysis, the average MUNE for the healthy side was significantly higher than the normal contralateral side only in the control group (TA-R vs. TA-L, 13.34 vs. 77.12; *p* < 0.05). On the other hand, in the scaffold group (TA-R vs. TA-L, 19.47 vs. 31.83), the treated side did not show significant differences compared to the muscle of the healthy limb (*p* > 0.05) (Figure 7).

The analysis of the ratio between the operated limb and the healthy limb at one year showed the following results:
Gastrocnemius muscle (MG-R vs. MG-L):
-In the study group (scaffold), the gastrocnemius muscle recovered 53.93% of its muscular function.-In the control group (suture), the gastrocnemius muscle recovered 35.13% of its muscular function (see Figure 8a).Tibialis anterior muscle (TA-R vs. TA-L):
-In the study group (scaffold), the tibialis anterior muscle showed a recovery of 61.15% of its muscular function.-In the control group (suture), the tibialis anterior muscle recovered 17.30% of its muscular function (see Figure 8b).

These results indicate that in the study group, with PBS scaffold, both the gastrocnemius and tibialis anterior muscles showed a higher percentage of recovery compared to the control group after one year.

The clinical observation of motility further allowed us to evaluate that all subjects treated with the scaffold showed a faster and more complete recovery of normal motor activity in the posterior limbs. Additionally, no fatalities were observed among the treated subjects (see Figure 9).

The present study has demonstrated that, in the long term, the use of a PBS scaffold is an effective surgical technique for improving nerve fiber regeneration in cases of complete peripheral nerve injuries in a murine model.

We believe that a microfibrillar PBS scaffold produced through electrospinning technique may represent a valid alternative to other types of scaffolds used. In 2018, Preethi Soundarya et al. [21] provided an overview of different fabrication techniques for scaffold preparation (biological macromolecules such as chitin/chitosan, collagen/gelatin, alginate, hyaluronic acid, silk, synthetic polymers, and ceramics). The study analyzed over 40 different techniques and compositions, showing promising results but concluding that an ideal fabrication method for a scaffold is yet to be defined. Actually, the PBS scaffold herein proposed is a viable alternative to epineurial suturing, currently considered the gold standard for nerve injuries.

A future objective will be to use growth factors in addition to the investigated scaffold to further enhance its demonstrated properties. In 2021, Ye et al. [22] demonstrated the effectiveness of a scaffold with continuous release of nerve growth factors in a murine model. Similarly, Gu X. et al. [23] developed a scaffold, referred to as TENG (Tissue Engineered Nerve Graft), supplemented with growth factors, that was capable of matching or even improving the outcome of autologous transplantation after nerve injuries.

## 4. Conclusions

This study in a mouse model showed that PBS scaffold repair of peripheral nerve lesions is superior to the gold standard technique of epineural suturing after one year.

Besides demonstrating the long-term biocompatibility of this material, histological examination showed regeneration with a greater number of fibers in nerve defects treated with scaffold implantation than in lesions treated with epineural suture alone. One year after implantation, histomorphological analysis revealed that in the population treated with epineurial suturing, the quantity and density of fibers downstream from the injury site were about 20% lower compared to the population treated with the PBS scaffold. Furthermore, the electromyographic evaluation demonstrated that the use of the scaffold in PBS resulted in a percentage recovery of muscular functionality approximately 20% higher than traditional sutures for the gastrocnemius muscle and over 45% higher for the tibialis anterior muscle.

These results highlight the potential of treating peripheral nerve injuries with the scaffold investigated in this study.

Therefore, considering that PBS scaffolds can be supplemented with growth factors, the results of this study represent only a part of the potential of these devices, encouraging future research on PBS scaffolds for the repair of peripheral nerve lesions.

## Figures and Tables

**Figure 1 polymers-15-03398-f001:**
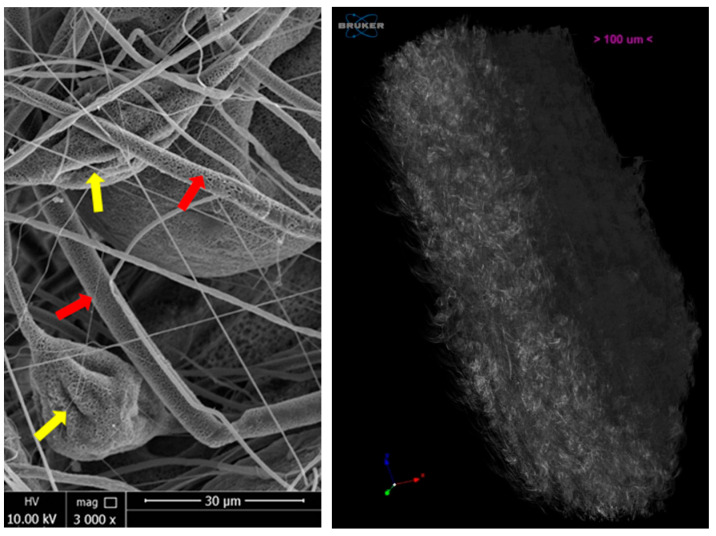
SEM (**left**) images at magnification 3000× (HV 10.00 kV; scale bar 30 micron) and micro-CT 3D reconstruction (**right**) of PBS scaffold. Micro-porosity on the fibers surface is indicated by yellow arrows. Microfibers are indicated by red arrows.

**Figure 2 polymers-15-03398-f002:**
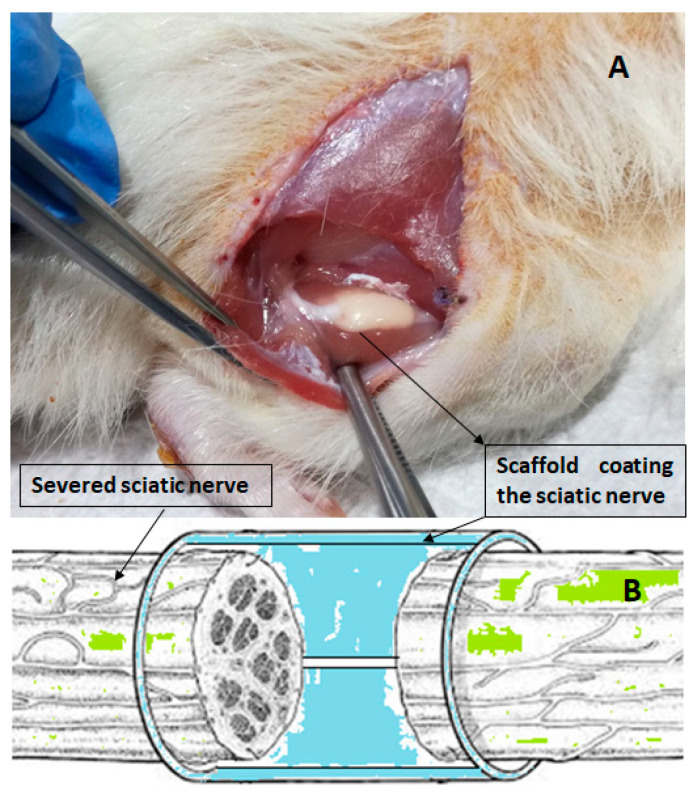
(**A**) Photo of the sciatic nerve wrapped with the scaffold, 3 weeks post implant. (**B**) Schematic representation of the scaffold guide action between the proximal and distal ends of the injured nerve.

**Figure 3 polymers-15-03398-f003:**
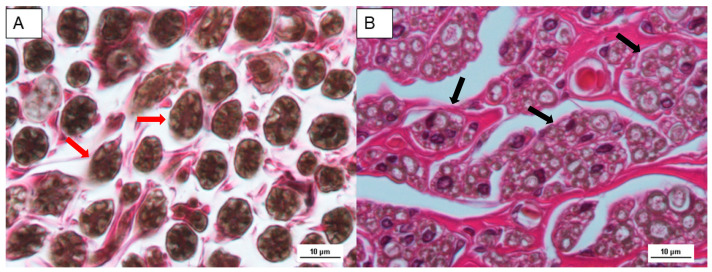
Histology images of Group 1 (**A**) and Group 2 (**B**) at 1 year post-surgery. Osmium Tetroxide fixation and hematoxylin-eosin stain. Scale bar 10 µm. Red arrows indicate large nerve fibers.

**Figure 4 polymers-15-03398-f004:**
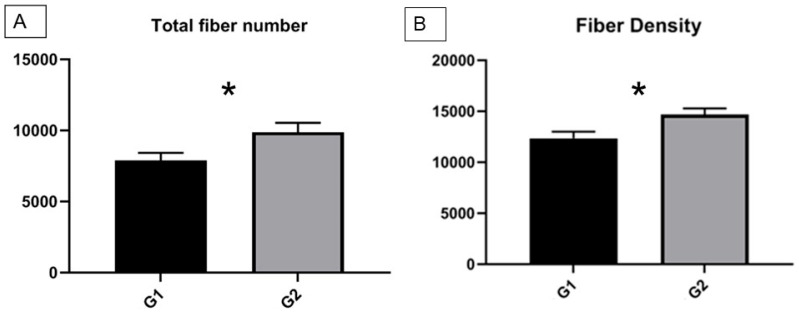
(**A**) Average number of total fibers after 1 year post-surgery in Group 1 (G1) and Group 2 (G2); (**B**) nerve fiber density after 1 year post-surgery. * *p* < 0.0001.

**Figure 5 polymers-15-03398-f005:**
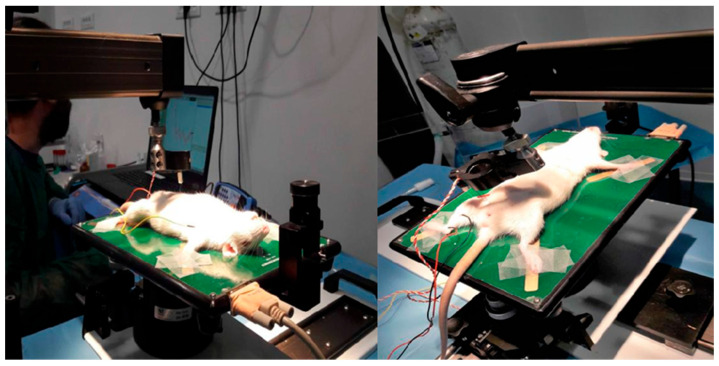
Experimental registration of CAMP.

**Figure 6 polymers-15-03398-f006:**
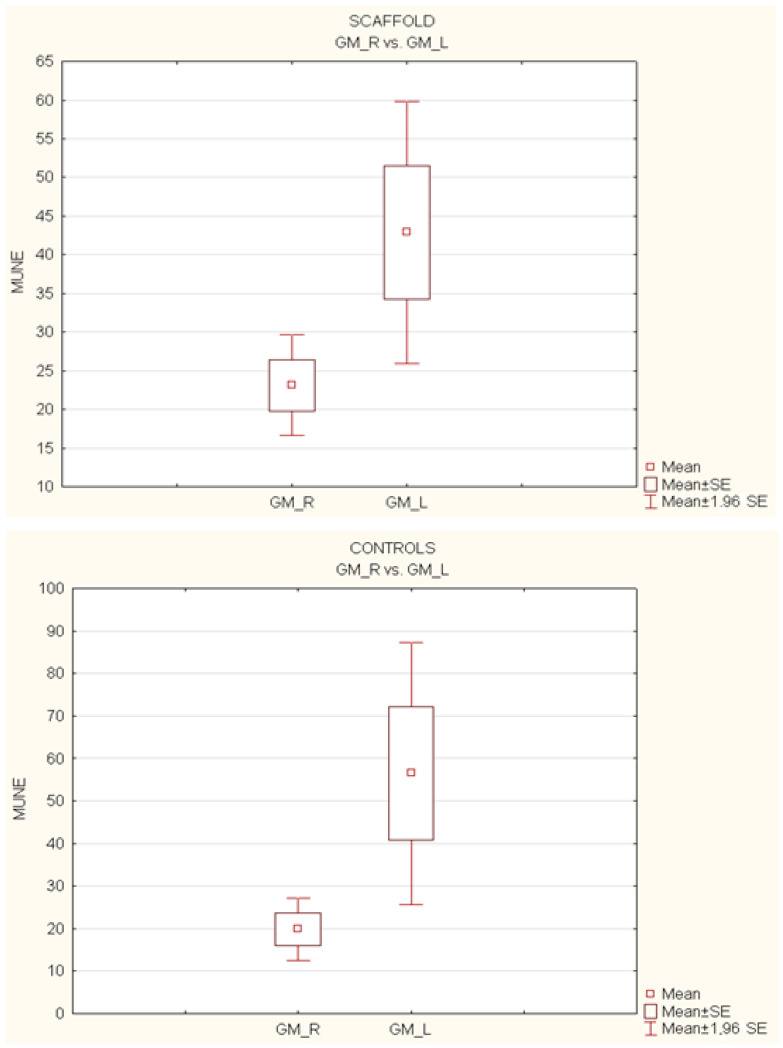
Comparison of mean MUNEs between operated (R) and non-operated (L) limbs for gastrocnemius (GM) muscles.

**Figure 7 polymers-15-03398-f007:**
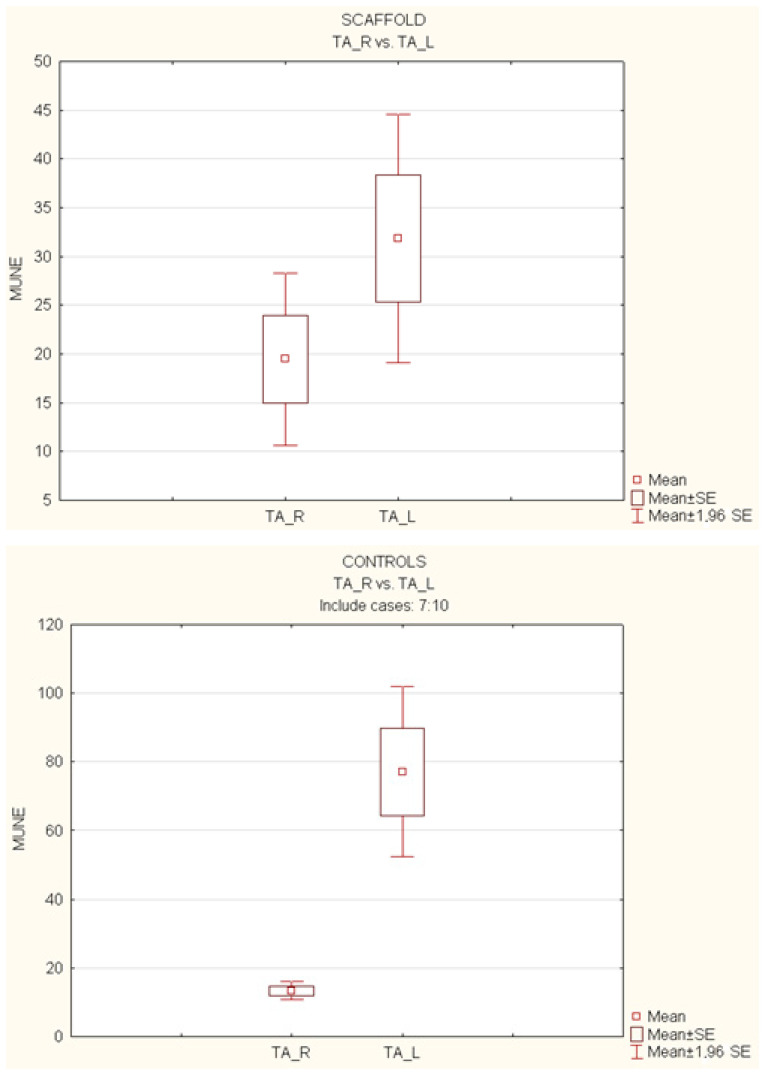
Comparison of mean MUNEs between operated (R) and non-operated (L) limbs for tibialis anterior (TA) muscles.

**Figure 8 polymers-15-03398-f008:**
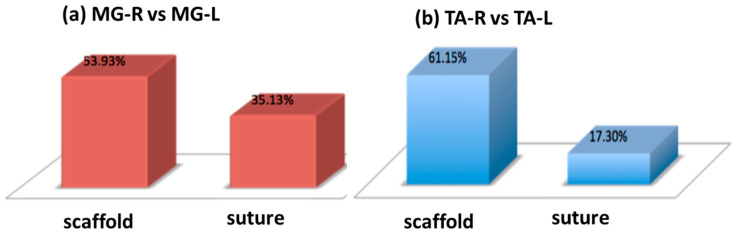
Graph of the percentage recovery of muscle function of the scaffold group vs. suture group in the gastrocnemius muscle (**a**) and tibialis anterior (**b**), respectively.

**Figure 9 polymers-15-03398-f009:**
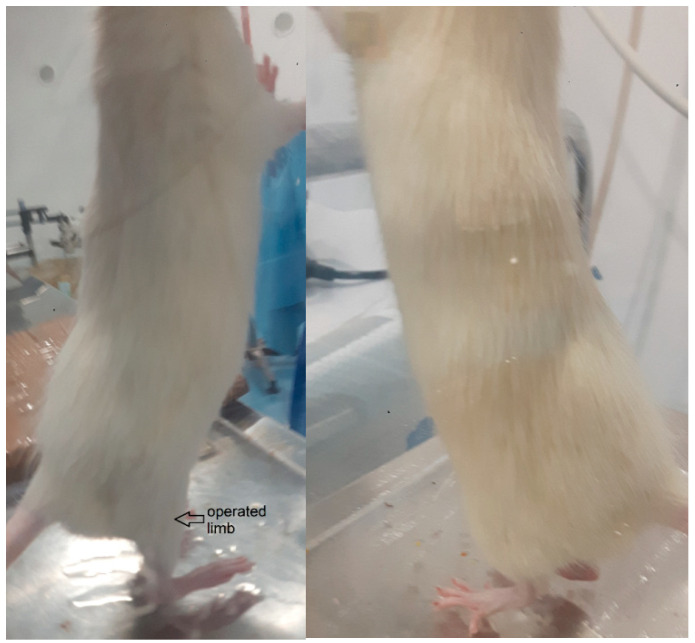
Photos of operated rats one year after implantation. Operated leg (**left**) vs. non operated leg (**right**).

## Data Availability

The data that support the findings of this study are available from the corresponding author upon reasonable request.

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
