# Peer review of "Peripheral Nerve Regeneration at 1 Year: Biodegradable Polybutylene Succinate Artificial Scaffold vs. Conventional Epineurial Sutures"

_polymers, 2023, doi:10.3390/polym15163398_

Round 1

Reviewer 1 Report

The manuscript was revised and the next should be explained.

The figures 1 and 3 should includea better description about observed. Please mark with arrows the observed.

The figure 4. The axis of X should be explained. 

Figure 5. Please explain the figure, why left and right side of the leg is showed ? 

Author Response

Dear Editor and reviewer,

the authors would like to thank you for the opportunity to submit a revised version of our manuscript 'Peripheral nerve regeneration at 1 year: biodegradable polybutylene succinate artificial scaffold vs. conventional epineurial sutures' (ID polymers-2541059).

We are grateful to the reviewers for their insightful comments on our article. We were able to incorporate the changes and suggestions provided by the reviewers. We have highlighted the changes within the manuscript in red.

Reviewer 1:

The figures 1 and 3 should include a better description about observed. Please mark with arrows the observed.

According to the reviewer's suggestion, Figures 1 and 3 were better described by adding new text and arrows in the figures. In Figure 3, a better description of the number and size of the fibres observed was added.

The figure 4. The axis of X should be explained. 

According to reviewer’s suggestion, X axis in figure 4 was explained in the legend of the figure.

Figure 5. Please explain the figure, why left and right side of the leg is showed? 

The two photos have been included to show that there is no difference between the operated limb (right) and the healthy limb (left). The latter has been indicated in the legend in Figure 9 of the revised manuscript.

Reviewer 2 Report

Optimized materials for peripheral nerve tissue repair are very important due to the low regeneration ability of the nerve. The author developed here the biodegradable polymers for peripheral nerve regeneration. The focus of this study is important and interesting, but the data was not enough to prove the concept. It would be better to indicate more additional support data and evaluation.

1.       Please explain the advantage of the planar structure materials for nerve tissue regeneration. What is the good point of using poly(1,4-butylene succinate)? Is it possible to prepare the graft by controlling the pore size and planar structure? The author should quantitatively indicate the physical properties and structural characteristics of the grafts.

2.       Since this journal is focused on the “Polymers”, basic information on physical properties such as the structure and molecular weight of the materials should be clarified. Please indicate this information in your manuscript.

3.       There were no control materials in this study, and it was hard to discuss the effectiveness of this material for nerve regeneration. Some other materials should be transplanted to the nerve in the same way and the number of regenerated axons should be compared to prove the concept.

4.       The author should indicate not only the HE staining but also immunostaining for Schwann cells and nerve cells in animal experimental results.  

5.       In the evaluation of peripheral nerve deficit in rats, nerve regeneration cannot be discussed only by macroscopic motility after 1 year.

6.       As the authors have pointed out, electrophysiological analysis is essential. CMAP data should be presented. A footprint evaluation would also be a general evaluation for nerve regeneration. Please consider this evaluation.

Author Response

Dear Editor and reviewer,

the authors would like to thank you for the opportunity to submit a revised version of our manuscript 'Peripheral nerve regeneration at 1 year: biodegradable polybutylene succinate artificial scaffold vs. conventional epineurial sutures' (ID polymers-2541059).

We are grateful to the reviewers for their insightful comments on our article. We were able to incorporate the changes and suggestions provided by the reviewers. We have highlighted the changes within the manuscript in red.

Reviewer 2:

Optimized materials for peripheral nerve tissue repair are very important due to the low regeneration ability of the nerve. The author developed here the biodegradable polymers for peripheral nerve regeneration. The focus of this study is important and interesting, but the data was not enough to prove the concept. It would be better to indicate more additional support data and evaluation.

  1. Please explain the advantage of the planar structure materials for nerve tissue regeneration. What is the good point of using poly(1,4-butylene succinate)? Is it possible to prepare the graft by controlling the pore size and planar structure? The author should quantitatively indicate the physical properties and structural characteristics of the grafts.

The advantage of using a planar scaffold for nerve regeneration was already showed in the previously published paper by Cicero et al.: Polybutylene succinate artificial scaffold for peripheral nerve regeneration, Journal of Biomedical Materials Research: Part B - Applied Biomaterials 110 (2022) 125–134. DOI: 10.1002/jbm.b.34896. Cicero al. demonstrated that the use of the planar PBS scaffold is an effective method to fix the two ends (proximal and distal) of the sciatic nerve injury, to reconstruct the continuity of the nerve and to promote its regeneration. In addition, the PBS scaffold is produced using the electrospinning technique. This technique allows to produce scaffolds with customised physical and structural properties, including porosity, size and shape (Polybutylene Succinate Processing and Evaluation as a Micro Fibrous Graft for Tissue Engineering Applications, G. C. Miceli, F. S. Palumbo , F. P. Bonomo, M. Zingales, M. Licciardi, Polymers (2022) 14(21), 4486. https://doi.org/10.3390/polym14214486). The authors believe that this material has great potential and hope that others will share our enthusiasm.

  1. Since this journal is focused on the “Polymers”, basic information on physical properties such as the structure and molecular weight of the materials should be clarified. Please indicate this information in your manuscript.

The authors thank the reviewer for this suggestion. Most of the structural characterisation is already reported in a previous published work (Polybutylene Succinate Processing and Evaluation as a Micro Fibrous Graft for Tissue Engineering Applications, G. C. Miceli, F. S. Palumbo , F. P. Bonomo, M. Zingales, M. Licciardi, Polymers (2022) 14(21), 4486. https://doi.org/10.3390/polym14214486). However, as suggested, the molecular weight characterization has been added in section 2.1 of the revised version of this manuscript.

  1. There were no control materials in this study, and it was hard to discuss the effectiveness of this material for nerve regeneration. Some other materials should be transplanted to the nerve in the same way and the number of regenerated axons should be compared to prove the concept.

Thank you for this suggestion. It would have been interesting to explore this aspect. However, our study focused on the development of this material and its comparison with a technique that is now the gold standard in surgery.

The authors agree that other materials could be used for nerve regeneration. Various natural polymers (e.g. collagen, silk fibroin, chitosan, alginate) or synthesised polymers (e.g. PU, PCL, PLA) have been successfully used in the manufacture of porous scaffolds by electrospinning. Although several electrospun polymers have been studied for peripheral nerve regeneration, none of them is PBS. The authors agree with the reviewer's assumption that a positive effect on nerve regeneration could also be expected from other materials. However, as PBS has not yet been studied, the authors decided to carry out this evaluation by comparing it primarily with suture repair. On the other hand, comparison with other commercial scaffolds or devices was not the aim of this study, which aims to demonstrate the potential of the new PBS polymer application. We do not exclude the possibility, in the future, of implementing our study by choosing a different control group, perhaps with scaffolds made from materials on the market and already studied.

  1. The author should indicate not only the HE staining but also immunostaining for Schwann cells and nerve cells in animal experimental results.

This point is certainly interesting. However, the authors considered post-fixation with osmium tetroxide to perform morphometric analyses. As reported in some scientific articles, this technique represents a fundamental investigative tool in the study of nerve regeneration. The following can be referred to in this connection: “A revisit to staining reagents for neuronal tissues” by A. Rosario, A. Howell, and S. K. Bhattacharya” in Ann Eye Sci. 2022 Mar; 7: 6; doi: 10.21037/aes-21-3. The authors considered that immunostaining may not be sufficiently accurate for our purpose.

  1. In the evaluation of peripheral nerve deficit in rats, nerve regeneration cannot be discussed only by macroscopic motility after 1 year.

The authors agree and have incorporated your suggestion to the revised manuscript. Please see the reply to point 6.

  1. As the authors have pointed out, electrophysiological analysis is essential. CMAP data should be presented. A footprint evaluation would also be a general evaluation for nerve regeneration. Please consider this evaluation.

The authors agree with the reviewer and themselves that "electrophysiological analysis is essential" to demonstrate the real effect of the scaffold on peripheral nerve regeneration. Actually, electromyographic analysis was included in the research project and carried out. Unfortunately, by the deadline for paper submission, the data from this analysis still needed to be reexamined, corrected, contextualized, and validated. Therefore, the authors deemed it appropriate to submit only the confirmed and validated data. We apologise for not including this information at submission, but doing so would not have been appropriate. After completing the analysis, the authors are in agreement with the reviewer and delighted to be able to add the electrophysiological analysis to the revised manuscript. Therefore, a paragraph on the electrophysiological evaluation and new figures were included to describe the obtained results.

Round 2

Reviewer 2 Report

This manuscript was appropriately revised follow by the reviewer's suggestion. The quality is sufficient for publication.